# Does Demineralized Bone Matrix Affect the Nonunion Rate in Arthroscopic Ankle Arthrodesis?

**DOI:** 10.3390/jcm11133893

**Published:** 2022-07-04

**Authors:** Carsten Schlickewei, Julie A. Neumann, Sinef Yarar-Schlickewei, Helge Riepenhof, Victor Valderrabano, Karl-Heinz Frosch, Alexej Barg

**Affiliations:** 1Department of Trauma and Orthopaedic Surgery, University Medical Center Hamburg-Eppendorf, Martinistraße 52, 20251 Hamburg, Germany; s.yarar@uke.de (S.Y.-S.); k.frosch@uke.de (K.-H.F.); al.barg@uke.de (A.B.); 2Department of Orthopaedics, University of Utah, 590 Wakara Way, Salt Lake City, UT 84108, USA; julie.neumann@hsc.utah.edu; 3Center for Trauma Rehabilitation and Sport Medicine, BG Hospital Hamburg, Bergedorfer Str. 10, 21033 Hamburg, Germany; helge.riepenhof@redbulls.com; 4Red Bull Athlete Performance Center, Brunnbachweg 71, 5303 Thalgau, Austria; 5Swiss Ortho Center, Schmerzklinik Basel, Swiss Medical Network, Hirschgässlein 15, 4010 Basel, Switzerland; vvalderrabano@swissmedical.net; 6Department of Trauma Surgery, Orthopaedics and Sports Traumatology, BG Hospital Hamburg, Bergedorfer Str. 10, 21033 Hamburg, Germany

**Keywords:** osteoarthritis: ankle fusion, ankle arthrodesis, arthroscopic, demineralized bone matrix, delayed union, nonunion

## Abstract

Demineralized bone matrix (DBM) has been shown to have positive effects on union rates in many orthopedic subspecialties; however, minimal evidence exists about bone graft substitutes in foot and ankle surgery. The purpose of this study is to compare nonunion rates in arthroscopic ankle arthrodesis in patients receiving DBM with those without. We hypothesized DBM to be associated with a decreased risk of nonunion. This retrospective review includes 516 consecutive ankle arthrodesis cases from March 2002 to May 2016. Of these, 58 ankles (56 patients) that underwent primary arthroscopic ankle arthrodesis met the inclusion criteria, and 31 of these ankles received DBM, while 27 did not. Nonunion was assessed by clinical examination and routine postoperative radiographs. If nonunion was suspected, a computed tomography (CT) scan was performed. The primary outcome measure was nonunion rate. Secondary outcome measures included wound complications, return to operating room (OR), and rate of postoperative deep vein thrombosis (DVT) or pulmonary embolism (PE). From the study cases, 58 were available for final follow-up. The average age was 55.9 years (±17.4), and mean follow-up was 43.0 months (range 6.3–119.4). There was no difference in nonunion rate in patients who received DBM (4/31, 12.9%) versus those who did not (4/27, 14.8%) (*p* = 0.83). Similarly, when comparing the two groups, there were no statistically significant differences in superficial wound complications (6.5% vs. 3.7%, *p* = 1.0) or rate of return to OR (29% or 0.037/person-years vs. 37% or 0.099/person-years; *p* = 0.20). No major complications including deep wound infections, DVTs, or PEs occurred. This is the largest study to directly compare nonunion rates and complications for patients receiving DBM versus those who did not in primary arthroscopic ankle arthrodesis. No significant association was found between DBM usage and risk of nonunion, wound complications, return to OR, or postoperative DVT or PE development.

## 1. Introduction

Ankle osteoarthritis is most commonly caused by previous trauma [1,2,3,4,5,6] and can result in significant pain and disability [7,8]. Nonoperative treatments, including activity modification, physical therapy, anti-inflammatories, shoe inserts/modifications, ankle braces, and tibiotalar joint corticosteroid injections, frequently do not provide lasting relief [4,9,10]. Thus, tibiotalar fusion is an established treatment option for patients suffering from end-stage arthritis. Open ankle arthrodesis has become the gold standard in operative treatment as it reliably alleviates joint pain and is able to correct deformities [4]. More recently, arthroscopic ankle arthrodesis has seen an increase in adoption as it avoids the morbidity associated with open procedures [11]. Many studies have examined the advantages of arthroscopic versus open ankle arthrodesis, with arthroscopic procedures showing lower complication rates, faster recoveries, and shorter hospital stays, among others [11].

However, nonunion is still one of the most common complications following open and arthroscopic arthrodesis, often resulting in revision surgery [12]. Nonunion rates have been reported to be as high as 40% [13,14,15,16,17]. Several non-modifiable risk factors predispose patients to nonunion, such as talar osteonecrosis, smoking, poor bone quality, diabetes mellitus, hemophilia, and inherent ankle deformity [18,19,20,21,22]. Rather than focusing on these often-unmodifiable risk factors, the attention has turned to perioperative modifications. This includes the use of bone grafts and bone graft substitutes, such as demineralized bone matrix (DBM), when bone grafting is needed to fill voids in arthrodesis sites [12]. DBM is a type of bone allograft that stimulates the induction of osteoblasts through proteinaceous growth factors [12]. Previous studies have already investigated the use of DBM in ankle arthrodesis; however, these studies were small and had multiple limitations.

This is currently the largest study evaluating the use of DBM in primary arthroscopic ankle arthrodesis. The purpose of this study is to evaluate the use of DBM and determine whether or not it affects nonunion rates in patients undergoing primary arthroscopic ankle arthrodesis.

## 2. Materials and Methods

### 2.1. Study Participants

For this study, 516 consecutive patients underwent ankle arthrodesis between March 2002 and May 2016, among whom 58 ankles from 56 patients who underwent primary arthroscopic ankle arthrodesis were included in this study. The inclusion criteria were severe tibiotalar arthritis (Takakura stage 2, 3, or 4) [23], failure of a minimum of three months of nonoperative treatment [4], ≥18 years of age, and a minimum of six months of radiographic follow-up [24]. All patients were treated by one of four fellowship-trained orthopedic foot and ankle surgeons. The exclusion criteria were open or revision tibiotalar arthrodesis or tibiotalar arthrodesis fixated with hardware other than screws. The data exclusions can be found in Figure 1. Of the final sample, 31 ankles had DBM placed in the tibiotalar arthrodesis site to stimulate fusion, and 27 procedures were performed without DBM. This study was conducted in accordance with the Declaration of Helsinki and the Guidelines for Good Clinical Practice. A retrospective analysis with fully anonymous clinical data was performed.

### 2.2. Surgical Technique

In all patients, arthroscopic ankle fusion was performed with the standard arthroscopic anteromedial and anterolateral portals [25,26] in combination with noninvasive distraction [27]. Care was taken to avoid the superficial peroneal nerve [28,29] when creating the anterolateral portal. Diagnostic arthroscopy was performed, paying particular attention to the condition of the talofibular joint. If the joint was well-maintained, the tibiofibular joint was preserved. If the joint demonstrated signs of osteoarthritis, a tibiofibular arthrodesis was performed through a separate lateral incision. The residual tibial and talar cartilage was removed with a combination of curettes, curved osteotomes, and arthroscopic shavers and burrs. The shapes of the tibial plafond and talar body were maintained if congruent. The lateral gutter (talofibular cartilage) was preserved if there were no plans to include it in the fusion. The tibia and talus were fenestrated using a 4–0 burr to induce stem cell egress into the joint (Figure 2a,b). The foot was positioned with slight dorsiflexion, external rotation, and valgus. Prior to compression, if demineralized bone matrix (DBX^®^ Inject™, Depuy Synthes, Solothurn, Switzerland) was utilized, 2–5 cc was inserted into the tibiotalar joint. There were no specific standardized indications for the use of DBM in this study. Initial fixation was obtained with a guidewire and verified fluoroscopically. If the position of the ankle was satisfactory, the tibiotalar joint was fixated with two or three 6.5 mm or 7.0 mm cannulated screws (Depuy Synthes or Medartis, Basel, Switzerland). If a tibiofibular arthrodesis was performed, two or more 3.5 mm non-cannulated screws were utilized to stabilize the distal tibiofibular arthrodesis. Bone graft that leaked out of the joint was irrigated. Final fluoroscopic imaging was obtained prior to wound closure. The wounds were then irrigated and closed with nylon sutures. All patients received postoperative splint or boot ankle immobilization and were non-weight-bearing for a minimum of six weeks. Generally, patients received follow-up at 2, 6, and 12 weeks, 6 months, and 1 year postoperatively (Figure 3).

### 2.3. Outcome Analysis

The primary outcome measure was the nonunion rate of primary arthroscopic ankle arthrodesis. Healing of the arthrodesis was defined by patient-reported symptoms and clinical examination criteria (stability with weight bearing, improvement in swelling, no warmth or pain). Radiological bone union was defined as visible trabecular bridging of at least 80% of the previous tibiotalar joint line in the coronal and sagittal views within six months after surgery [30]. Nonunion was assessed by routine postoperative radiographs, and if delayed union or nonunion was suspected, a computed tomography (CT) scan was performed. Secondary outcome measures included the need for revision surgery, wound complications, and rate of postoperative deep vein thrombosis (DVT) or pulmonary embolism (PE).

### 2.4. Statistical Analysis

Patient demographics and clinical characteristics were summarized descriptively for all tibiotalar arthrodesis cases and stratified by DBM use. Normally distributed continuous variables were summarized using means and standard deviations (SD), and the two groups were compared using a *t*-test. Non-Gaussian distributed variables were summarized as median and interquartile range (IQR) and compared using the Wilcoxon rank-sum test. Categorical variables were summarized as frequencies and percentages and compared using a chi-squared test or Fisher’s exact test.

The primary outcome, including the nonunion rate of those with DBM usage and those without, was calculated using a univariable logistic regression and unadjusted odds ratio. Statistics included 95% confidence intervals (CI) and *p*-values. Secondary outcomes were compared using Fisher’s exact tests, and rate of return to the operating room (OR) was compared using a log-rank test. Since the rate of return to OR depended on patient follow-up, we summarized rate of return stratified by DBM status using person-years. Statistical significance was defined as *p* < 0.05, and all tests were two-sided. Statistical analysis was performed using IBM SPSS Statistics version 26.0 (IBM, Armonk, NY, USA).

## 3. Results

### 3.1. Demographics and Clinical Characteristics

For this study, 58 ankles (56 patients) were available for a mean final follow-up of 43.0 months (range 6.3–119.4). Demographic and clinical characteristics are presented in Table 1. There were no statistically significant differences in preoperative deformity in those who received versus those who did not receive DBM (Table 2). The ankle arthritis etiology is described in Table 3.

### 3.2. Postoperative Outcomes/Complications

The nonunion rate for patients receiving DBM was 4/31 (12.9%), and it was 4/27 (14.8%) for those who did not (*p* = 0.83) (Table 4). The odds ratio for nonunion in those with DBM was 0.9 (95% CI: 0.2–4.0, *p* = 0.83). There were no statistically significant differences between the two groups when comparing superficial wound complications (6.5% vs. 3.7%, *p* = 1.00) and rate of return to OR (29% or 0.04/person-years vs. 37% or 0.1/person-years; *p* = 0.20). None of the patients who had superficial wound infections required operative debridement. There were no major complications in this study, including deep wound infections, DVT, or PE.

## 4. Discussion

This study demonstrates comparable rates of union as well as postoperative complications (including nonunion, wound complications, rate of return to OR, and thromboembolic events) in patients who received DBM versus those who did not.

There have been a handful of other studies examining the utilization of DBM in the foot and ankle. First, in 1996, Crosby et al. evaluated 41 patients after arthroscopic ankle arthrodesis with a bi-framed distraction technique [31]. At an average follow-up of 27 months, 85% of patients were satisfied. This study reported a 7% nonunion rate and ultimately concluded that DBM did not increase the union rate. However, this study is a case series of only 41 patients and did not compare a cohort of patients who received DBM with those who did not [31].

Second, in 1996, Michelson and Curl evaluated 55 patients who underwent either a triple arthrodesis or subtalar fusion with the addition of either an iliac crest bone graft or DBM [32]. This study showed no difference in time to union or union rate between those who received iliac crest autograft or DBM. The authors concluded that DBM assists with hindfoot arthrodesis at least as well as iliac crest autograft, without the additional blood loss and postoperative pain seen in iliac crest grafting. This study compared iliac crest autograft to DBM; however, the tibiotalar arthrodesis procedure was not examined [32].

Iliac crest autograft is the current gold standard, as it provides live cells and is both osteoinductive and osteoconductive. Despite that, it has high rates of morbidity and incurs more expenses when compared with DBM [12,32]. Advantages of DBM include that it is osteoinductive, circumventing the need to harvest bone from a separate surgical site, and that it has been shown to result in timely healing without increasing complication rates [33]. However, it should be noted that commercial demineralization processes may vary widely, and some of the procedures to attenuate residual pathogens and antigens can cause damage to the graft itself. [33,34,35] Additionally, different brands of DBM may have different efficacies in vivo on the union of arthrodesis sites.

Third, in 2003, Thordarson and Kuehn published a case series of 63 patients who underwent complex ankle or hindfoot fusion that utilized DBM (37 Grafton putty, 26 Orthoblast) [12]. They showed that 14% (5/37) of patients who underwent arthrodesis with Grafton ^®^ DBM putty (BioHorizons^®^, Birmingham, AL, USA) experienced nonunion versus 8% (2/26) of patients with Orthoblast^®^ (Integra, Irvine, CA, USA). Ultimately, it was concluded that there was no difference in nonunion rate between those two particular products. Likewise, the authors determined there to be no difference in union rate when compared with historical controls, although there was no control group in this study [12]. Additionally, only 6 of the 63 patients received tibiotalar arthrodesis [12].

Fourth, in 2006, Collman et al. published a retrospective study that included 39 patients who underwent arthroscopic ankle arthrodesis by a single podiatric surgeon between 1994 and 2003 [19]. The mean age was 65 years, and patients were noted to have minimal to no ankle deformity. At one-year follow-up, the authors noted an 87.2% (34/39) union rate with a statistically significantly decreased rate in obese patients. Collman et al. demonstrated that neither DBM nor platelet-rich plasma (PRP) improved ankle arthrodesis union rates. However, this study is underpowered: Only 7 of 39 patients received DBM, and thus it is hard to draw conclusions [19].

In our study, the nonunion rate was 13.8%, which is slightly higher than what has previously been reported for ankle arthrodesis procedures (approximately 10%) [12,15,31,36]. We attribute this to the complexity of the patient population treated at our university medical center. Additionally, the rate of return to OR in this study is high, 32.5%. We are aggressive with the removal of painful/prominent hardware, and 19% of patients returning to the OR did so for hardware removal. This study demonstrated a 12.1% rate of revision arthrodesis, as seven of the eight patients with nonunion returned to the OR. This is consistent with previous literature, showing a high rate of revision arthrodesis in the setting of nonunion [16].

Limitations of this study include its retrospective nature, although all hindfoot surgical data were collected in a prospective manner into the databank. Second, the amount of DBM utilized, surgical technique, and postoperative protocols were not standardized, as four different fellowship-trained orthopedic foot and ankle surgeons treated these patients. Additionally, of the patients who did not receive DBM, 4/27 (14.8%) had proximal tibial autograft, and 1/27 (3.7%) had iliac crest autograft placed within the tibiotalar arthrodesis, which may have confounded the results. However, this supports the conclusion that DBM does not decrease nonunion rates, as the non-DBM ankle union rate may have been positively influenced by the use of autografts. Additionally, there was a significantly higher percentage of tibiofibular fusions in ankles that used DBM, which may or may not have affected the nonunion rate. The effect of tibiofibular joint inclusion in ankle arthrodesis has not been examined in the literature. Lastly, concomitant procedures that may have affected nonunion rates were not recorded.

To our knowledge, this is the largest study directly comparing nonunion rates and complications in ankles that underwent primary arthroscopic tibiotalar arthrodesis with and without the use of DBM. This study suggests that the use of DBM does not decrease the rate of nonunion, nor does it affect the rate of return to OR or major complications including deep wound infection, DVT, and PE.

## 5. Conclusions

This is the largest study directly comparing nonunion rates and complications in patients who underwent primary arthroscopic ankle arthrodesis with and without the use of DBM. No significant associations were found between the use of DBM and the risks of nonunion, wound complications, return to the OR, or development of postoperative DVT or PE in arthroscopic tibiotalar arthrodesis. According to our data, the use of DBM in ankles undergoing primary arthroscopic tibiotalar arthrodesis is a safe procedure and not associated with an increased rate of nonunion or complications. Further randomized prospective studies with larger patient populations are needed to conclusively clarify the advantages and disadvantages of DBM use in primary arthroscopic ankle arthrodesis.

## Figures and Tables

**Figure 1 jcm-11-03893-f001:**
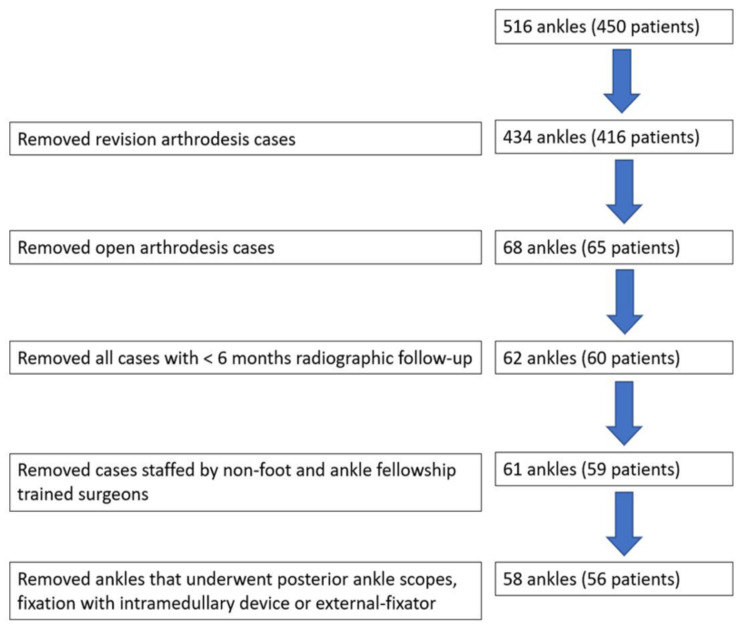
The data exclusion flow chart for the present study.

**Figure 2 jcm-11-03893-f002:**
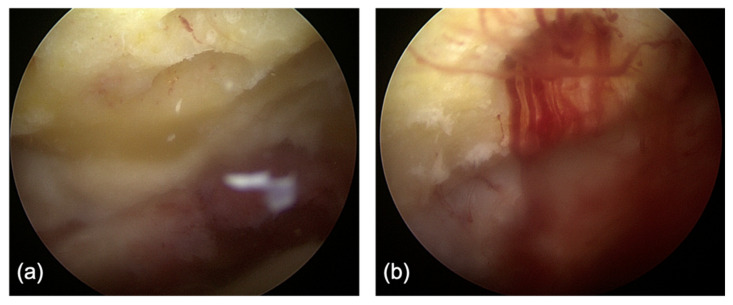
View from the anterolateral arthroscopic portal of a right ankle showing the fenestration of the subchondral bone on the tibial side (**a**) to induce stem cell ingress into the tibiotalar joint. The same view (**b**) after tourniquet removal, demonstrating bleeding bone.

**Figure 3 jcm-11-03893-f003:**
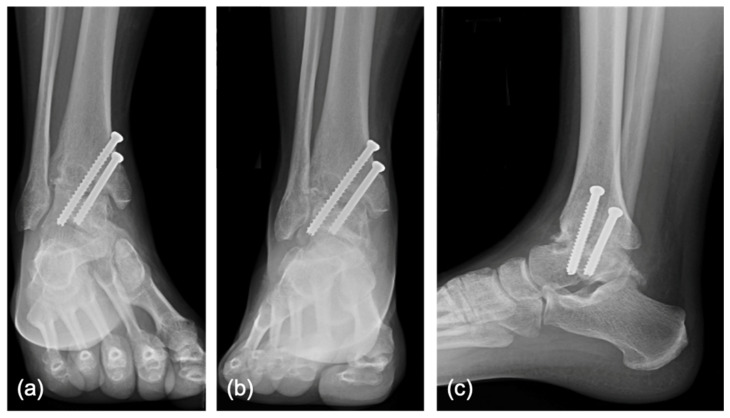
Immediate postoperative radiographs, including non-weight-bearing mortise (**a**), anterior–posterior (**b**), and lateral (**c**) views, of a 26-year-old male with hemophilia who underwent arthroscopic tibiotalar arthrodesis of the right ankle with the supplementation of demineralized bone matrix.

**Table 1 jcm-11-03893-t001:** Demographics and clinical characteristics.

Variable	All Ankles (*n* = 58)	Ankles with DBM (*n* = 31)	Ankles without DBM (*n* = 27)	*p*-Value
Age [years ± SD]	55.9 ± 17.4	58.2 ± 18.5	53.2 ± 15.9	0.28 ^†^
Gender (female)	33 (56.9%)	21 (67.7%)	12 (44.4%)	0.07 ^‡^
BMI with range [kg/m^2^]	29.2 (27.0–31.6)	28.7 (25.1–31.5)	29.6 (27.9–33.3)	0.11 *
Diabetes	7 (12.1%)	3 (9.7%)	4 (14.8%)	0.69 ^‡^
Smokers	3 (5.2%)	1 (3.2%)	2 (7.4%)	0.59 ^‡^
Right-sided surgery	27 (46.6%)	16 (51.6%)	11 (40.7%)	0.41 ^‡^

^†^ *t*-test; ^‡^ Chi-square test; * Wilcoxon rank-sum test; BMI = Body mass index; SD = Standard deviation.

**Table 2 jcm-11-03893-t002:** Preoperative deformity measures.

Preoperative Deformity	All Ankles (*n* = 58)	Ankles with DBM (*n* = 31)	Ankles without DBM (*n* = 27)	*p*-Value
MDTA with range [°]	88° (86–90°)	88° (86–90.5°)	89° (87–90°)	0.39 ^†^
TTT with range [°]	0° (−2–0°)	0° (−3–0°)	0° (−0.8–1°)	0.07 ^†^
CMA [°] with range	−4.1° (−11.5–2.1°)	−8.8° (−11.8–3.1°)	−3.6° (−6.3–0.3°)	0.47 ^†^
ADTA [°] with range	83° (80–85°)	83° (80.5–85°)	83° (78.5–85.5°)	0.50 ^†^

^†^ Wilcoxon rank-sum test; ADTA = Anterior distal tibial angle; CMA = Calcaneal moment arm (negative values indicate varus malalignment); MDTA = Medial distal tibial angle; TTT = Tibiotalar test.

**Table 3 jcm-11-03893-t003:** Etiology of tibiotalar osteoarthritis.

Variable	All Ankles (*n* = 58)	Ankles with DBM (*n* = 31)	Ankles without DBM (*n* = 27)	*p*-Value
Primary	7 (12.1%)	3 (9.7%)	4 (14.8%)	0.69 ^†^
Secondary (including posttraumatic OA)	51 (87.9%)	28 (90.3%)	23 (85.2%)	

^†^ Fisher’s exact test; OA = Osteoarthritis.

**Table 4 jcm-11-03893-t004:** Postoperative complications.

Variable	All Ankles (*n* = 58)	Ankles with DBM (*n* = 31)	Ankles without DBM (*n* = 27)	*p*-Value
Nonunion	8 (13.8%)	4 (12.9%)	4 (14.8%)	0.83 ^†^
Superficial wound complications	3 (5.2%)	2 (6.5%)	1 (3.7%)	1.00 ^‡^
Return to OR	19 (32.8%, 0.06/person-year)	9 (29.0%, 0.04/person-year)	10 (37.0%, 0.1/person-year)	0.20 ^‡^
• Peroneus brevis tear	1 (3.7%)	0 (0%)	1 (3.7%)	
• Revision arthrodesis	7 (12.1%)	3 (9.7%)	4 (14.8%)	
• ROH	11 (19%)	6 (19.4%)	5 (18.5%)	

^†^ Based on a univariable logistic regression; ^‡^ Chi-square test; OR = Operating room; ROH = Removal of hardware.

## Data Availability

Derived data supporting the findings of this study are available from the author upon reasonable request.

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
