# Peer review of "Does Demineralized Bone Matrix Affect the Nonunion Rate in Arthroscopic Ankle Arthrodesis?"

_jcm, 2022, doi:10.3390/jcm11133893_

Round 1

Reviewer 1 Report

The manuscript is interesting with regard to the topic addressed, with an overall case history that is not so small.

The methodological approach is correct and the conclusions are in line with the results obtained.

It could be worthy of publication, also because it is very relevant to the topic of the special issue, however an extensive linguistic revision by a native English or Editing system is a priority (there are many grammatical errors in the manuscript, the English used is not always scientifically correct).

Also, probably wrong in all tables is the number of Ankles without DBM indicated as '31', instead it should be '27'.

In line 53, diabetes and haemophilia could be added among the risk factors (also considering that the authors presented a case of haemophilia in their case history), if useful I recommend the following citations:

- Greco T, Polichetti C, Cannella A, La Vergata V, Maccauro G, Perisano C. Ankle hemophilic arthropathy: literature review. Am J Blood Res. 2021 Jun 15;11(3):206-216. PMID: 34322283; PMCID: PMC8303020.

- Cianni L, Bocchi MB, Vitiello R, Greco T, De Marco D, Masci G, Maccauro G, Pitocco D, Perisano C. Arthrodesis in the Charcot foot: a systematic review. Orthop Rev (Pavia). 2020 Jun 25;12(Suppl 1):8670. doi: 10.4081/or.2020.8670. PMID: 32913602; PMCID: PMC7459387.

Author Response

Dear Reviewer,

thank you very much for your review.

We have taken your comments into consideration and incorporated them into the text.

  1. In accordance with your justified comment regarding the written language and style, extensive linguistic revision has been made by a native English speaker.

(the manuscript contains many grammatical errors, the English used is not always scientifically correct).

In this regard, the entire text has been completely reviewed and revised once in terms of written language, style and scientific correctness.

  1. The data in the text and tables were checked again individually and the error that you noticed was corrected.

(wrong in all tables is the number of Ankles without DBM indicated as '31', instead it should be '27').

Thank you for your attention!

We apologize for this mistake!

  1. As you suggested, I have added in line 59/60, diabetes and haemophilia and also incorporated the two references you recommended.

(diabetes and haemophilia could be added among the risk factors (also considering that the authors presented a case of haemophilia in their case history), if useful I recommend the following citations:).

Unfortunately, although I would like to, I cannot follow your suggestion to present a case of hemophilia in our case history.

Tragically, our friend and senior author, Prof. Barg, passed away suddenly in mid-April. The data and research were conducted under his direction and also archived by him. This article was the last one he released for publication before his sudden death. Unfortunately, due to his death, we no longer have access to all image files. It is therefore not possible for us to present a case of hemophilia in our case history.

Under these circumstances, I hope you agree that we cannot include a corresponding case in our case presentation and understand the exceptional situation we are in.

Dear reviewer,

after revision of this paper and based on the fact that this paper fits well into the special issue "Ankle Osteoarthritis" and is currently the largest study investigating primary arthroscopic ankle arthrodesis with or without DBM, we ask you to review the paper again and consider it for publication.

Thank you and best regards

Carsten Schlickewei

Reviewer 2 Report

Dear Schlickewei et al.,

The manuscript “Does Demineralized Bone Matrix Affect the Nonunion Rate in Arthroscopic Ankle Arthrodesis?” (jcm-1739642) by Schlickewei et al. evaluate the use of the DBM and to determine whether or not it affects the nonunion rate in patients undergoing primary arthroscopic ankle arthrodesis.. The topic is interesting, but I think this article should reconsider after proper changes in major revision for publication in Journal of Clinical Medicine. Some of my specific comments are below:

1. Describe the novelty of the article made by the author? From the results of my evaluation, it seems that many similar published works adequately explain what you have raised in the current manuscript. If there are something others really new in this manuscript, please highlight it more clearly in the introduction section (line 40-61).

2. The state of the art and the significance of the current study are not clearly present, the authors should highlight it more advanced in the introduction section (line 40-61).

3. In the introduction section (line 30-85), the authors should explain the previous research conducted and its shortcomings. It will uphold the research gap that you filled with your research novelty. I recommend the authors elaborate on their introduction section. Do not forget to attention carefully to my previous comments on numbers 1 and 2.

4. Since this manuscript evaluates medical implant in a clinical perspective, discussion from a computational perspective needs to be explained for more comprehensive explanation. Also, I would encourage and advise the authors to adopt some of the specific additional references related to medical implant from the computational simulation perspective published by MDPI in the introduction and/or discussion section as follow:

Tresca Stress Simulation of Metal-on-Metal Total Hip Arthroplasty during Normal Walking Activity. Materials (Basel). 2021, 14, 7554. https://doi.org/10.3390/ma14247554

The Effect of Bottom Profile Dimples on the Femoral Head on Wear in Metal-on-Metal Total Hip Arthroplasty. Journal of Functional Biomaterials. 2021, 12, 38. https://doi.org/10.3390/jfb12020038

Computational Contact Pressure Prediction of CoCrMo, SS 316L and Ti6Al4V Femoral Head against UHMWPE Acetabular Cup under Gait Cycle. Journal of Functional Biomaterials. 2022, 13, 64. https://doi.org/10.3390/jfb13020064

5. Figure 2 (line 105) and Figure 5 (line 110) needs to be arranged to fit into one page only without being divided into the other page like the present form.

6. Standard/basis/protocol for outcome analysis (line 114-121) needs to be explained.

7. A more clear explanation/step-by-step statistical analysis (line 122-137) is needed to detailed explanation in the present manuscript.

8. The author must provide a detailed specification and use condition more detail regarding all tools used in the research carried out so that the reader can estimate the accuracy and differences in the results that the authors describe due to the use of different tools in future studies.

9. The sample used in the present manuscript is relatively small that would brings to bias results and interpretation. There is any explanation that supports the use of a small group in the conducted research since the sample use impacts the overall information to the reader?

10. Patient-involved detail is not clearly present. It is important since every patient have personalized result and impact the result, such as activity level, origin, height, body mass index, and other. To avoid the misinterprestation, patient detail needs more explainded. In table 1 (line 145), this information is not enough.

11. In the Results and discussion section (line 138-165), the authors are advised to compare the results they obtain with previous similar/identical studies if it is possible.

12. In the last paragraph before conclusion section (after line 236), the authors should add of one paragraph about the limitations of the presented study.

13. The conclusion (line 237-242) of the present manuscript is not solid. Further elaboration is needed.

14. Further research needs to be explained in the conclusion section (line 237-242).

15. Overall, the quality of the present article is not giving a significant scientific contribution, major improvement is needed in the quality and quantity aspects. Extending explanation with comprehensive discussion is also important.

16. In the whole of the manuscript, the authors sometimes made a paragraph only consisting of one or two sentences that made the explanation not clearly understood. The authors need to extend their explanation to become a more comprehensive paragraph. In one paragraph, it is recommended to consist of at least 3 sentences with 1 sentence as the main sentence and the other sentences as supporting sentences. For example in line 167-169.

17. I see some errors on English in some areas of the present manuscript. To improve the quality of English used in this manuscript and make sure English language, grammar, punctuation, spelling, and overall style are correct, further proofreading is needed. As an alternative, the authors can use the MDPI English proofreading service for this issue.

18. Please make sure the authors have used the Journal of Clinical Medicine, MDPI format correctly. The authors can download published manuscripts by Journal of Clinical Medicine, MDPI, and compare them with the present author's manuscript to ensure typesetting is appropriate. For example: Athors conrtribution, Funding, Institutional Review Broad Statement, Informed Consent Statement, Data Availability Statement, and Conflict of Interest in line 243-248 is not filled by the authors.

I am pleased to have been able to review the author's present manuscript. Hopefully, the author can revise the current manuscript as well as possible so that it becomes even better. Good luck for the author's work and effort.

Best regards,

The Reviewer

Author Response

Dear Reviewer,

thank you very much for your time and your detailed review. We greatly appreciate your commitments and support to improve the quality and scientific relevance of this paper.

I have discussed your recommendations with the co-authors and we have tried to incorporate all your suggestions for improvement into the paper.

Tragically, our friend and senior author, Prof. Barg, passed away suddenly in mid-April. The data and research were conducted under his direction and also archived by him. This article was the last one he released for publication before his sudden death. Unfortunately, his death has made it difficult for us to implement some of your suggestions for improvement.

Nevertheless, we have made every effort to comply with your recommendations.

I hope you agree with our improvements and understand the exceptional situation we are in.

In response to your recommendations in detail:

  1. Describe the novelty of the article made by the author? From the results of my evaluation, it seems that many similar published works adequately explain what you have raised in the current manuscript. If there are something others really new in this manuscript, please highlight it more clearly in the introduction section (line 40-61).
  2. The state of the art and the significance of the current study are not clearly present, the authors should highlight it more advanced in the introduction section (line 40-61).
  3. In the introduction section (line 30-85), the authors should explain the previous research conducted and its shortcomings. It will uphold the research gap that you filled with your research novelty. I recommend the authors elaborate on their introduction section. Do not forget to attention carefully to my previous comments on numbers 1 and 2. 

1.-3.

We understand your objection. As you wrote, there are already studies that have investigated or described this connection. We explicitly point this out in the discussion section. (There have been a handful of other studies on the utilization of DBM in the foot and ankle...) The studies are cited individually and their weaknesses are discussed in detail.

- 1996, Crosby et al - this study is a case series of only 41 patients and did not compare the cohort to patients that underwent arthroscopic ankle fusions without DBM

- 1996, Michelson and Curl compared iliac crest autograft to DBM, but it should be noted that no tibiotalar arthrodesis were included

- 2003, Thordarson and Kuehn published historical controls, but they did not have a control group in this study. In this study, only 6 of the 63 patients were tibiotalar arthrodesis.

- 2006, Collman et al published a retrospective study - only 7 of 39 patients had DBM thus it is hard to draw conclusions

This work is currently the largest study evaluating primary arthroscopic ankle arthrodesis with or without DBM.

We have included a reference to this in the Introduction section, as you recommended.

In addition, we have tried to better highlight the value of arthroscopic ankle arthrodesis. Many studies have investigated the advantages of arthroscopic ankle arthrodesis over open surgery, implying an advantage such as lower complication rate, faster recovery and shorter hospital stay.

We have additionally supplemented the introduction with regard to known risk factors for nonunion and substantiated these with 2 additional references [21 and 22]. There are a number of known risk factors that are not modifiable that predispose patients to nonunion such as osteonecrosis of the talus, smoking, poor bone quality, diabetes, haemophilia and inherent ankle deformity [18-22].

Based on your recommendation, we now explicitly point out, that previous studies regarding DBM and ankle arthrodesis exist, but these were small and had multiple limitations influencing the validity of the results.

Previous studies already investigated the use of DBM in ankle arthrodesis. However, these studies were small and had multiple limitations.

  1. Since this manuscript evaluates medical implant in a clinical perspective, discussion from a computational perspective needs to be explained for more comprehensive explanation. Also, I would encourage and advise the authors to adopt some of the specific additional references related to medical implant from the computational simulation perspective published by MDPI in the introduction and/or discussion section as follow:

Tresca Stress Simulation of Metal-on-Metal Total Hip Arthroplasty during Normal Walking Activity. Materials (Basel). 2021, 14, 7554. https://doi.org/10.3390/ma14247554

The Effect of Bottom Profile Dimples on the Femoral Head on Wear in Metal-on-Metal Total Hip Arthroplasty. Journal of Functional Biomaterials. 2021, 12, 38. https://doi.org/10.3390/jfb12020038

Computational Contact Pressure Prediction of CoCrMo, SS 316L and Ti6Al4V Femoral Head against UHMWPE Acetabular Cup under Gait Cycle. Journal of Functional Biomaterials. 2022, 13, 64. https://doi.org/10.3390/jfb13020064

4.

The literature you recommended to us is very interesting. The studies are very meaningful, especially with regard to total hip arthroplasty. Unfortunately, it was not possible for us to add the literature from the computational simulation perspective specifically in context, or to insert it on the subject of arthroscopic ankle arthrodesis or bone substitute materials (DBM).

We apologize for that!

  1. Figure 2 (line 105) and Figure 5 (line 110) needs to be arranged to fit into one page only without being divided into the other page like the present form.

5.

The figures have been rearranged, cropped and labeled in the text.

  1. Standard/basis/protocol for outcome analysis (line 114-121) needs to be explained.
  2. A more clear explanation/step-by-step statistical analysis (line 125-135) is needed to detailed explanation in the present manuscript.
  3. The author must provide a detailed specification and use condition more detail regarding all tools used in the research carried out so that the reader can estimate the accuracy and differences in the results that the authors describe due to the use of different tools in future studies.

6.-8.

Outcome analysis section and statistical analysis section have been expanded and clarified to more clearly define the primary and secondary outcome measures (line 126-136) as well as the statistical analysis (line 138-153).

The statistical analysis was done by Prof. Barg himself, as in almost all his publications, as he was very structured and meticulous here and was very reluctant to let this out of his hands in order to avoid errors. Outcome Analysis section and Statistical Analysis section were written by him and approved for publication. We have now tried to further specify the data analysis.

We apologize for not being able to ask Prof. Barg personally for cooperation in this matter.

Statistical analysis and outcome analysis were supplemented and completed. Specification and execution were done according to previous publications in this journal. All instruments used in the research conducted were provided in the text.

  1. The sample used in the present manuscript is relatively small that would bring to bias results and interpretation. There is any explanation that supports the use of a small group in the conducted research since the sample use impacts the overall information to the reader? 

9.

You are right, the sample used in this manuscript is relatively small.

Nevertheless, this work is currently the largest study evaluating primary arthroscopic ankle arthrodesis with or without DBM.

We have included a reference to this in the Introduction section, as you recommended.

Based on your recommendation, we now explicitly point out, that previous studies regarding DBM and ankle arthrodesis exist, but these were even smaller and had multiple limitations influencing the validity of the results.

  1. Patient-involved detail is not clearly present. It is important since every patient have personalized result and impact the result, such as activity level, origin, height, body mass index, and other. To avoid the misinterprestation, patient detail needs more explainded. In table 1 (line 145), this information is not enough.

10.

This study is an anonymized retrospective study. All available data in this regard were evaluated.

The data and research were conducted under the direction of Prof. Barg and were also archived by him. A subsequent extension of the data evaluation is unfortunately excluded due to his death.

  1. 11. In the Results and discussion section (line 138-165), the authors are advised to compare the results they obtain with previous similar/identical studies if it is possible.

11.

In accordance with your recommendation, a detailed discussion of previous studies is provided in line 188 to line 231, outlining the imitations and the results compared to this study.

  1. In the last paragraph before conclusion section (after line 236), the authors should add of one paragraph about the limitations of the presented study.

12.

Based on your recommendations, the Limitations section (lines 232-245) has been significantly expanded to point out all limitations of the study.

  1. The conclusion (line 237-242) of the present manuscript is not solid. Further elaboration is needed.
  2. Further research needs to be explained in the conclusion section (line 237-242).

13.-14.

Conclusion was expanded and further research were added and specified.

  1. Overall, the quality of the present article is not giving a significant scientific contribution, major improvement is needed in the quality and quantity aspects. Extending explanation with comprehensive discussion is also important.

15.

In accordance with your reasoned comments on the quality and scientific contribution, we have attempted to significantly improve the paper. it is very relevant to the topic of the special issue, 

As part of this, a comprehensive linguistic revision of the written language and style was also carried out by a native English speaker

  1. In the whole of the manuscript, the authors sometimes made a paragraph only consisting of one or two sentences that made the explanation not clearly understood. The authors need to extend their explanation to become a more comprehensive paragraph. In one paragraph, it is recommended to consist of at least 3 sentences with 1 sentence as the main sentence and the other sentences as supporting sentences. For example in line 167-169.
  2. I see some errors on English in some areas of the present manuscript. To improve the quality of English used in this manuscript and make sure English language, grammar, punctuation, spelling, and overall style are correct, further proofreading is needed. As an alternative, the authors can use the MDPI English proofreading service for this issue.

16.-17.

In accordance with your justified comments regarding the written language and style, extensive linguistic revision has been made by a native English speaker.

In this regard, the entire text has been completely reviewed and revised once in terms of written language, style and scientific correctness.

  1. Please make sure the authors have used the Journal of Clinical Medicine, MDPI format correctly. The authors can download published manuscripts by Journal of Clinical Medicine, MDPI, and compare them with the present author's manuscript to ensure typesetting is appropriate. For example: Athors conrtribution, Funding, Institutional Review Broad Statement, Informed Consent Statement, Data Availability Statement, and Conflict of Interest in line 243-248 is not filled by the authors.

18.

This section has been completely revised and the relevant data inserted in the text.

Dear Reviewer,

Thank you for your time and efforts to significantly improve this paper through your comments.

In accordance with your reasoned comments on the quality and scientific contribution, we have tried to improve the paper significantly and to highlight the relevance of the paper to the topic of this Special Issue.

As part of this, a comprehensive linguistic revision of the written language and style was also carried out by a native English speaker

After extensive revision of this paper and based on the fact that this paper fits well into this Special Issue "Ankle Osteoarthritis" and is currently the largest study investigating primary arthroscopic ankle arthrodesis with or without DBM, we ask you to review the paper again and consider it for publication.

Thank you and best regards

Carsten Schlickewei

Round 2

Reviewer 1 Report

I greatly appreciate the efforts made by the authors for the detailed and thorough review of the article at such a difficult time for them.

The article, thanks to the point-by-point revision and language editing, in its current form is worthy of publication.

Reviewer 2 Report

Thank you to the author for his efforts in answering everything I criticized and making improvements to the manuscript. Regarding the death of one of the authors in this manuscript, I am deeply saddened. Personally, I also appreciate the writer's hard work to make the script better. Unfortunately, there are still many shortcomings in the current manuscript, even some things the author is not able to address properly. With all due respect, I recommend this manuscript be rejected and not published. It is very dangerous to accept manuscripts that are distorted and unclear.